# Targeted Next-Generation Sequencing in a Large Cohort of Genetically Undiagnosed Patients with Neuromuscular Disorders in Spain

**DOI:** 10.3390/genes11050539

**Published:** 2020-05-11

**Authors:** Lidia Gonzalez-Quereda, Maria Jose Rodriguez, Jordi Diaz-Manera, Jorge Alonso-Perez, Eduard Gallardo, Andres Nascimento, Carlos Ortez, Daniel Natera-de Benito, Montse Olive, Laura Gonzalez-Mera, Adolfo Lopez de Munain, Miren Zulaica, Juan Jose Poza, Ivonne Jerico, Laura Torne, Pau Riera, Jose Milisenda, Aurora Sanchez, Gloria Garrabou, Isabel Llano, Marcos Madruga-Garrido, Pia Gallano

**Affiliations:** 1Genetics Dept. Hospital de Sant Pau, IIB Sant Pau, 08041 Barcelona, Spain; MRodriguezF@santpau.cat (M.J.R.); PRiera@santpau.cat (P.R.); pgallano@santpau.cat (P.G.); 2U705, U762, U703, 722 and GCV4 for Biomedical Research on Rare Diseases (CIBERER), Instituto de Salud Carlos III, 28029 Madrid, Spain; Jordi.Diaz-Manera@newcastle.ac.uk (J.D.-M.); EGallardo@santpau.cat (E.G.); anascimento@sjdhospitalbarcelona.org (A.N.); dnatera@sjdhospitalbarcelona.org (D.N.-d.B.); GARRABOU@clinic.cat (G.G.); isabel.llano@osakidetza.eus (I.L.); 3Neuromuscular Unit, Neurology Dept., Hospital de Sant Pau, IIB Sant Pau, 08041 Barcelona, Spain; JAlonsoP@santpau.cat; 4Neuromuscular Unit, Neuropaediatrics Department, Hospital Sant Joan de Déu, Institut de Recerca Sant Joan de Déu, 08950 Barcelona, Spain; ciortez@sjdhospitalbarcelona.org; 5Neuropathology Unit, Department of Pathology and Neuromuscular Unit, Department of Neurology, IDIBELL-Hospital de Bellvitge, Hospitalet de Llobregat, 08907 Barcelona, Spain; molive@bellvitgehospital.cat (M.O.); 36747lgm@comb.cat (L.G.-M.); 6Department of Neurology, Hospital de Viladecans, 08840 Barcelona, Spain; 7Biodonostia, Neurosciences Area, Neuromuscular diseases Laboratory, San Sebastian, 20014 Basque Country, Spain; ADOLFOJOSE.LOPEZDEMUNAINARREGUI@osakidetza.eus (A.L.d.M.); MIREN.ZULAICAIJURCO@osakidetza.eus (M.Z.); 8CIBERNED, Instituto de Salud Carlos III, Ministry of Science, Innovation and Universities, 28029 Madrid, Spain; 9Department of Neurology, Hospital Universitario Donostia, San Sebastian, 20014 Basque Country, Spain; JUANJOSE.POZAALDEA@osakidetza.eus; 10Department of Neurosciences, Faculty of Medicine and Dentistry, UPV-EHU, San Sebastian, 48940 Basque Country, Spain; 11Navarre Institute for Health Research (IdiSNA), 31008 Pamplona, Spain; ivonne.jerico.pascual@navarra.es (I.J.); laura.torne.hernandez@navarra.es (L.T.); 12Department of Neurology, Complejo Hospitalario de Navarra, 31008 Pamplona, Spain; 13Hospital Clinic de Barcelona and Universidad de Barcelona, 08036 Barcelona, Spain; JCMILISE@clinic.cat; 14Department of Biochemistry and Molecular Genetics, Hospital Clinic de Barcelona, 08036 Barcelona, Spain; ASANCHEZ@clinic.cat; 15Cellex, IDIBAPS, University of Barcelona-Hospital Clínic of Barcelona, 08036 Barcelona, Spain; 16Biocruces Bizkaia Health Research Institute, Barakaldo, 48903 Bizkaia, Spain; 17Genetics Service, Cruces University Hospital, Osakidetza Basque Health Service, Barakaldo, 48903 Bizkaia, Spain; 18Instituto de Biomedicina de Sevilla (IBiS), Hospital Universitario Virgen del Rocío/CSIC, Universidad de Sevilla, 41013 Sevilla, Spain; mmadruga@us.es; 19Neuromuscular Disorder Unit, Pediatric Neurology Department, Hospital Universitario Virgen del Rocío, 41013 Sevilla, Spain

**Keywords:** neuromuscular diseases, congenital myopathies, muscular dystrophies, congenital myasthenic syndromes, targeted next-generation sequencing

## Abstract

The term neuromuscular disorder (NMD) includes many genetic and acquired diseases and differential diagnosis can be challenging. Next-generation sequencing (NGS) is especially useful in this setting given the large number of possible candidate genes, the clinical, pathological, and genetic heterogeneity, the absence of an established genotype-phenotype correlation, and the exceptionally large size of some causative genes such as *TTN*, *NEB* and *RYR1.* We evaluated the diagnostic value of a custom targeted next-generation sequencing gene panel to study the mutational spectrum of a subset of NMD patients in Spain. In an NMD cohort of 207 patients with congenital myopathies, distal myopathies, congenital and adult-onset muscular dystrophies, and congenital myasthenic syndromes, we detected causative mutations in 102 patients (49.3%), involving 42 NMD-related genes. The most common causative genes, *TTN and RYR1*, accounted for almost 30% of cases. Thirty-two of the 207 patients (15.4%) carried variants of uncertain significance or had an unidentified second mutation to explain the genetic cause of the disease. In the remaining 73 patients (35.3%), no candidate variant was identified. In combination with patients’ clinical and myopathological data, the custom gene panel designed in our lab proved to be a powerful tool to diagnose patients with myopathies, muscular dystrophies and congenital myasthenic syndromes. Targeted NGS approaches enable a rapid and cost-effective analysis of NMD- related genes, offering reliable results in a short time and relegating invasive techniques to a second tier.

## 1. Introduction

Neuromuscular disorders (NMD) is an umbrella term for genetic and acquired diseases that have traditionally been classified as primary disorders of the muscle (myopathies and muscular dystrophies), neuromuscular transmission disorders (myasthenia and myasthenic syndromes), or primary disorders of the peripheral motor neuron (spinal muscular atrophies and neuropathies). Onset may occur in the neonatal period, in childhood, or in adult life. Many NMDs have a genetic origin and around 600 genes involving more than a thousand neuromuscular diseases were described to date [1].

Myopathy and muscular dystrophy are broad terms that encompass many diseases in which the muscle fibers do not function properly. These diseases are usually characterized by degeneration of the skeletal muscles or structural abnormalities of the muscle fiber causing generalized muscle weakness and motor disability [2]. Although clinical trials are being carried out with promising results for some conditions, most disorders do not yet have any effective treatment [3]. Muscle biopsy findings are extremely useful to classify and differentiate these disorders. Muscular dystrophies (MD), limb girdle muscular dystrophies (LGMD) and congenital muscular dystrophies (CMD) typically show a dystrophic pathological pattern and some can be characterized by immunohistochemical and/or biochemical techniques [4]. In contrast, congenital myopathies (CM) present specific structural changes in the muscle fiber, such as cores, nemaline bodies and central nuclei [5], distinguished by histochemistry and electron microscopy. 

Congenital myasthenic syndromes (CMS) are a group of genetic diseases caused by abnormal signal transmission between motor axons and skeletal muscle fibers. These disorders are caused by molecular defects in genes coding for proteins that are located at the neuromuscular junction. Symptoms vary from mild to severe, but generally include muscle weakness and fatigability with or without ptosis and ophthalmoparesis. They can be misdiagnosed as muscular dystrophies or congenital myopathies. Determining the molecular defect is particularly relevant in treatment decision-making because a drug may be effective, ineffective, or even harmful depending on the type of CMS [6,7]. Precise genetic diagnosis in NMDs is therefore crucial to establish a prognosis, to better manage the disease, to offer genetic counseling to families, and to gain access to gene therapy trials. 

Mitochondrial diseases partially overlap with clinical presentation of neuromuscular disorders. Despite mitochondriopathies constitute a heterogeneous group of diseases with widely varying clinical features, involvement of both muscle and nerve is quite common, due to its dependence on mitochondrial activity. Consequently, myopathy and neuropathy are a major, often presenting, feature of several mitochondrial syndromes. In addition, mitochondrial dysfunction may play a role in several classic neuromuscular diseases. However, clinical guidelines and protocols were developed to discriminate between mitochondrial and neuromuscular disorders that genetically rely on different etiology [8,9].

The molecular diagnosis of inherited skeletal muscle diseases is a challenge not only due to their high clinical and genetic heterogeneity but also to the large number and great diversity of genes involved. As a reference laboratory that receives samples from all over the country, for many years we have conducted genetic studies, sequencing many types of muscular dystrophies gene by gene. Before next-generation sequencing (NGS) techniques became available, we were unable to address the pooled analysis of CM, CMD, MD and CMS. 

In view of the clinical, pathological, and genetic heterogeneity of NMDs, the relative inconsistence of the genotype-phenotype correlation, and the exceptionally large size of some of the causative genes, such as *TTN*, *NEB*, *RYR1* and *DMD*, NGS is particularly useful and appropriate to achieve a molecular diagnosis. Targeted resequencing of genes of interest is thus becoming widely used in the field, showing high efficiency and cost-effectiveness for routine diagnostics [10,11,12]. Furthermore, this technique is being applied as a first-tier test in the genetic screening of patients with a suspected neuromuscular disorder [13,14,15,16,17]. 

Our aim was to evaluate a custom targeted resequencing neuromuscular gene panel as a diagnostic tool to characterize the spectrum of congenital myopathies, congenital muscular dystrophies, muscular dystrophies and congenital myasthenic syndromes from the genetic point of view in the Spanish population. We also expected to establish a feasible and efficient workflow to better manage and diagnose NMD patients.

## 2. Materials and Methods 

### 2.1. Patients

We included 207 index patients (119 males and 88 females) with NMDs. Onset of the disease ranged from prenatal and congenital onset to late onset (Appendix A). All patients were followed up at 20 neurology and neuropediatric departments throughout Spain. Patients were suspected of having congenital myopathy (CM), congenital muscular dystrophy (CMD), muscular dystrophy (MD) (including all forms of LGMDs), or congenital myasthenic syndrome (CMS). Clinical suspicion was based on clinical signs or symptoms of muscle weakness, fatigability, or an elevated level of serum creatine kinase (CK), or on pathological features in muscle biopsy, when available. To be more concrete, 29 of the 207 patients had a clinical diagnosis of CM, including centronuclear and nemaline myopathy (in seven patients). Sixteen patients were referred as a CMD, including *LAMA2* (6 patients) and *COL6*-related CMD (9 patients). Eighteen patients were referred with the clinical suspicion of LGMD, and one patient was referred with suspected plectinopathy. CMS was suspected in 19 patients. The remaining patients were referred with a suspected diagnosis of myopathy. 

Patients suspected of having spinal muscular atrophy, facioscapulohumeral muscular dystrophy, or myotonic muscular dystrophy were not included because targeted NGS is not the most appropriate diagnostic technique in such cases. In addition, patients suspected of having mitochondrial disease were not included in this study. We also excluded patients with suspected dystrophinopathies because a particular test and workflow for Duchenne and Becker muscular dystrophies diagnosis is performed in our lab. However, *DMD* was also included in the custom gene panel used for this study so that we would be able to detect patients and female carriers whose clinical or pathological features did not raise suspicion of Duchenne or Becker Muscular Dystrophy No other inclusion or exclusion criteria were considered to select or exclude patients as we wanted to evaluate the diagnostic power of this tool in a real context of genetic diagnosis. Not all the enrolled patients (89 out of 207; 43%) had had a muscle biopsy before the genetic testing. 

Written informed consent was obtained from all the subjects. In the case of patients under 18 years, written informed consent was authorized and signed by parents or guardians. The study was approved by the Ethics Committee at Hospital de la Santa Creu i Sant Pau (HSCSP) and was performed in accordance with the ethical principles of the Helsinki Declaration (WMA, 2013) 

### 2.2. Genetic Analysis

We selected 116 genes known to cause skeletal muscle disorders (congenital myopathies, congenital and limb girdle muscular dystrophies, congenital myasthenic syndromes and non-dystrophic myotonias (Table 1)) to design a Nextera Rapid Capture panel (Illumina) using the Design Studio Sequencing tool (https://designstudio.illumina.com/) (Illumina, San Diego, CA, USA). 

Targets included coding regions with 25 bp of flanking intronic sequences. The resulting custom Nextera Rapid Capture assay covered 99% of selected regions comprising a cumulative target length of 635,221 base pair (bp) with the design of 4058 probes and an average fragments size of 200 bp. The concentration of genomic DNA was determined using the Qubit^®^ dsDNA BR Assay kit for use with the Qubit 2.0 Fluorometer (ThermoFisher Scientific, Waltham, MA, USA). A total of 50 ng of genomic DNA were used to construct the library. DNA was enzymatically fragmented and library amplifications, hybridizations and purifications were performed following the manufacturer’s instructions (Illumina, San Diego, CA, USA). Libraries were quantified using the dsDNA BR Assay kit and equimolarly pooled to achieve an equal representation of all the samples. The final library was diluted to 4nM and loaded onto MiSeq at a concentration of 12pM. Sequencing was performed using the MiSeq Platform with the MiSeq Reagent Kit v2 (300-cycles) cartridge to obtain 150 bp paired-end reads and an average coverage of at least 100× of the target region. MiSeq Reporter software (Illumina, San Diego, CA, USA) was used to perform secondary data analyses and to process base calls generated on-instrument during the sequencing run. This software supplies information about alignment, variants, and contiguous assemblies for each sample. We used the Integrative Genomic Viewer (IGV, Broad Institute and UC San Diego, CA, USA) for visualization and interactive exploration of Binary Alignment Map (BAM) files.

Messenger RNA analysis was performed by cDNA sequencing. Total mRNA was extracted and purified from approximately 30 mg of muscle using RNeasy Fibrous Tissue Mini Kit (Qiagen, Hilden, Germany) and subsequently retrotranscribed to cDNA by RT-PCR using polythymine primers (ThermoFisher Scientific, Waltham, MA, USA). Self-designed primers were used to amplify the cDNA region in which the intronic mutation was located including two 5′ and 3′ contiguous exons. 

### 2.3. Data Analysis and Interpretation

Variant annotation and filtering were undertaken using Illumina Variant Studio 3.0 (Illumina), starting with Variant Call Format (VCF) files from each sample. Variants had to meet two requirements to remain on the list and not be filtered out. The first requirement was a read depth above 30× and the second was a population variant frequency reported in Thousand Genomes, Exome Aggregation Consortium (ExAC), Genome Aggregation Database (gnomAD) and Exome Variant Server (EVS) below 5%. Variant analyses and their interpretation were performed using Alamut Visual v2.6 software (Interactive Biosoftware, Rouen, France) to assess the impact of missense mutations and the variant’s effect on splicing predictions. The software also gathered information coming from public databases such as gnomAD, ESP, Cosmic, and ClinVar. Variants were prioritized by: (i) high quality scores and coverage; (ii) truncating variants as nonsense, frameshift and splicing sites, and missense variants located at functional domains; and (iii) genotype-phenotype correlation. Synonymous variants with in silico prediction of some impact on splicing were also selected. Variant pathogenicity was assessed according to the American College of Medical Genetics (ACMG) guidelines [18]. Once candidate variants were selected in each case, the results were discussed in multidisciplinary groups formed by a clinician expert in neuromuscular diseases, a pathologist and a geneticist to determine the cause of disease. Clinically relevant variants were confirmed by Sanger sequencing. This technique was useful to confirm the presence of variants, to rule out technical artifacts, and to exclude crossing samples when preparing libraries. Additionally, we performed an MLPA (Multiplex Ligation-dependent Probe Amplification) directed to *CAPN3* gene in patients in whom a single *CAPN3* mutation was identified and a high genotype-phenotype correlation was found. The SALSA MLPA P176 CAPN3 probe mix (MRC-Holland, Netherlands) was used following the manufacturer’s instructions. We also performed an MLPA using SALSA MLPA P309 MTM1 probe mix in a patient in whom a deletion in *MTM1* was suspected by coverage analysis of NGS data. Furthermore, when feasible, we analyzed selected variants at mRNA level and/or segregation analyses in families. 

## 3. Results

### 3.1. Coverage and Sequencing Depths

The mean sequencing depth of all samples was 211.8× and uniformity of coverage was 94.9%. The target average coverage was 96.5% at 20× (the coverage of sites with depth greater than 20 in the target region) and 91.2% at 50×. The average sequencing depth of all samples was 219.71, with a standard deviation of 97.20.

### 3.2. Identified Variants 

Causative mutations were detected in 102 of the 207 patients (49.3%), involving 42 different NMD-relates genes. No disease-causing mutation was identified in 73 patients (35.3%). The remaining 32 cases (15.4%) were partially solved. These cases corresponded to patients carrying variants of uncertain significance (VUS) that closely matched the phenotype and patients with suspected recessive diseases lacking a second mutation to fully explain their disease. Results of the genetic analyses in each studied patient are available in the Appendix A. 

The most common causative genes identified were *RYR1* (16 out of 102 cases; 15.7%), *TTN* (14 cases; 13.7%), *COL6A1* (5 cases; 4.9%), *MYOT*, *DES* and *ANO5* (4 cases each gene; 3.9% each) and *LAMA2*, *DNM2*, *CHRNE* and *ACTA1* (3 cases each gene; 2.9% each). The remaining causative genes were present in one or two cases (Figure 1).

All but one of the *RYR1* mutations identified in our cohort were missense variants. The exception was a nonsense mutation c.9157 C > T, located at exon 61, not previously described. The most prevalent phenotype associated with the *RYR1* mutations identified in this work was central core myopathy in both dominant and recessive forms of disease inheritance (Appendix A).

Twenty-four patients showed *TTN* candidate variants, 14 of them achieving a confirmed genetic diagnosis due to either segregation analysis was available or they carried a truncating variant. Thirteen patients showed a single variant, and 11 patients showed two variants. Twenty-five variants were missense, six were frameshift, three were nonsense, and one was a splicing variant. We identified the Iberian mutation c.107889delA [19] in two patients: P32 and P173. The most prevalent phenotypes associated with *TTN* mutations were distal muscle weakness and anterior tibial weakness, mainly inherited in a dominant manner (Appendix A). 

Among the 29 patients that were referred with a suspected CM, 20 (69%) were confirmed with a genetic diagnosis. The CM causative genes were *MTM1*, *RYR1*, *DNM2*, *CHRNA1*, *ACTA1*, *TTN*, *NEB* and *COLQ* genes. Nine of 16 patients (56%) in whom a CMD was suspected were genetically diagnosed with mutations identified in *LAMA2*, *COL6A1*, *COL6A2*, *COL6A3* and *GMPPB* genes, and two remained as partially solved since a second mutation in *LAMA2* and *COL6A1* genes was missing. Finally, 79% of patients in whom a LGMD was suspected obtained a genetic diagnosis with identified mutations in *CAPN3*, *RYR1*, *CHRND*, *FKRP*, *TTN*, *DES*, *DYS*, *MYOT*, *TNPO3*, *PLEC* and *ANO5* genes (Appendix A). 

Congenital myasthenic syndromes were genetically diagnosed in 16 patients (7.7%). Six of them (P29, P50, P124, P158, P186, P198) were referred as atypical forms of muscular dystrophies and/or myopathies. Patient P29, a 49-year-old male with no family history of NMD, was referred to our laboratory with the diagnosis of a centronuclear myopathy. He had been examined in two different centers when he was 7 and 19 years old due to proximal muscle weakness in upper and lower limbs. A muscle biopsy performed when the patient was 21 showed moderate unspecific myopathic changes. A further muscle biopsy performed when he was 48 years showed 80% of internal nuclei (Appendix A). His sequencing data showed a known and reported *COLQ* mutation, c.1289A > C (rs121908923), in homozygous state [20] (Appendix A). In view of this finding, and in absence of other strong candidate variants, the patient underwent an electromyography study (EMG) that showed a decremental response to repetitive nerve stimulation, confirming the identified *COLQ* variant as the cause of his disease. 

If we look only at patients in whom a definite genetic cause was identified, 87 had mutations in a heterozygous state (54 with a single heterozygous mutation and 33 with a compound heterozygous state) while 15 patients were homozygous for the mutations they carried. Eight patients were born from consanguineous parents. We were able to count 101 alleles carrying a missense mutation, 24 alleles with a frameshift mutation, 9 alleles harboring a nonsense mutation, 3 alleles carrying an in frame small deletion or duplication, and 12 alleles showing a splicing mutation. All splicing variants were located at +1, −1 and +2 splice site positions, except for patients P82 and P118, who showed different intronic location variants. P82 showed the *COL6A1* c.717 + 4A > G variant as one of the only candidate variants that correlated with the patient’s phenotype. This variant was selected after checking the LOVD database (www.dmd.nl), which provided mRNA analysis of the mentioned variant, demonstrating that it produces a transcript with a frameshift p. [=, Ile239fs ∗ 30]. Patient P118 harboured two variants in *DOK7*, the common c.1124_1127dup mutation in the maternal allele, and an intronic short deletion c.54 + 11_54 + 22del in the paternal allele that had also been reported as pathogenic [21]. The patient improved with ephedrine.

An autosomal-dominant inheritance pattern was observed in 52 families and an autosomal recessive inheritance pattern was found in 48 families. Two families showed ×-linked inheritance. In one family, an adult male (P98) harbored a missense mutation located at the exon 6 of *DMD* gene, which contains an acting binding site. After this finding, the patient was re-examined showing a phenotype compatible with a Becker muscular dystrophy that was confirmed by a reduction of dystrophin in the muscle biopsy immunostaining. *DMD* deletions and duplications were discarded by MLPA analysis. In the other family, a young boy (P35) had a deletion from exon 7 to 15 in *MTM1* gene. This deletion was confirmed by MLPA (Appendix A).

The dominant or recessive inheritance of the *RYR1*, *TTN* and *COL6* variants was established, in some cases, by checking previous reports and through variant segregation in the family when available. We were able to segregate the mutation in 49 families, 15 of whom were shown to carry de novo dominant mutations. The inheritance pattern remained uncertain in 18 patients (P4, P8, P10, P20, P24, P38, P64, P68, P106, P109, P111, P120, P146, P170, P183, P194, P202, and P205). This was because a heterozygous variant with a strong phenotype correlation was identified in a gene (mainly *TTN*, *RYR1, COL6A1,* and *CAPN3*) that can cause both autosomal-dominant and autosomal recessive diseases. We were unable to perform segregation analysis to clarify the inheritance in these families due to unavailability of samples from relatives. 

Fifteen patients carried variants of unknown significance. All of these were missense changes that correlated with the patients’ phenotype but required additional tests, such as segregation analysis or functional studies, to confirm their causality. This VUS group also included three patients (P8, P24 and P146) in whom only a heterozygous *CAPN3* mutation was identified, correlating with the patients’ phenotype. Regarding “partially solved cases”, patient 4 (P4) presented a frameshift mutation and a missense variant in the *DAG1* gene that correlated perfectly with the patient’s phenotype. When these variants were segregated in the family, we found that they were inherited in a *cis* manner. In the case of patients P7, P11, P15, P37 and P140, we were unable to identify the second causative mutation (Appendix A). 

## 4. Discussion

In the present study, we implemented a diagnostic strategy within a routine genetic diagnostic procedure by using a custom next-generation sequencing-based panel directed to several neuromuscular diseases. As far as we know, a comprehensive and integrative genetic diagnostic method of congenital and adult myopathies, muscular dystrophies and myasthenic syndromes has not been previously performed in Spain.

In the era of Sanger sequencing it was often difficult to establish a genetic diagnosis of these entities because many large and complex genes were involved. Stored samples from a large number of adults and children with NMDs samples therefore awaited genetic diagnoses. We received candidate samples from medical centers all over the country but a detailed clinical history, a muscle biopsy, and relatives willing to undergo a genetic study were not always available. 

In the present study, despite the lack of clinical information in some cases, genetic diagnosis was conclusive in 102 out of 207 patients (49.3%), involving 42 different NMD-related genes. Our diagnostic yield is consistent with that referred to in the bibliography, which is generally between 40% and 60% [22,23,24,25,26].

The most frequently mutated gene in our cohort was *RYR1*, identified in 14.5% of patients, closely followed by *TTN,* which was the causal gene in 13.7% of patients. Together, *RYR1* and *TTN* account for almost 30% of solved and partially solved cases in which the genetic cause was identified or strongly suspected. Sequencing throughput in the study of neuromuscular disorders was limited in terms of time and cost-efficiency with the Sanger method, especially when large genes were involved. *RYR1* and *TTN* therefore clearly benefit from high throughput technologies such as *NGS* and are proving to be a major cause in myopathies. As Davis MR et al. and Amburgey K et al. stated *RYR1* mutations are preferentially grouped in three hotspots: an N-terminal region including exons 2 to 7, a central region comprising exons 39 to 46, and a C-terminal region encompassing exons 85 to 103 [27,28]. In our cohort, most *RYR1* mutations were missense variants located at central and C-terminal hotspots. Of particular note, we found that many *RYR1* mutations were concentrated in exons 101, 102 and 103. Curiously, we observed that all identified dominant *RYR1* mutations were positioned on a hotspot. When looking at recessive cases, we found at least one of the causative mutations was also placed in a hotspot of the *RYR1* gene. 

The second most frequently mutated gene, *TTN,* showed diverse clinical manifestations, ranging from congenital onset presenting type 1 fibre predominance to late onset and mild muscle weakness, with central nuclei as a pathological sign on muscle biopsy (Appendix A). Like Savarese and colleagues, we also observed that most *TTN* mutations were located in the distal part of the gene [29]. Based on our variant segregation results, it seems that solved cases of *TTN* have a predominance of a recessive inheritance pattern (11 *TTN* recessive cases out of 15 solved *TTN* cases) but we should be cautious here due to the difficulty in assessing and classifying *TTN* missense variants. In all but one case, patients who presented symptoms during childhood had recessive inheritance. The exception was patient P170 in whom we only identified a nonsense mutation. Since this patient presented congenital hypotonia, it remained uncertain whether the *TTN* mutation was causative of the disease or whether a second *TTN* mutation was not identified. Single *TTN* missense variants identified in patients for whom no relatives’ samples were available for segregation analysis remain as *VUS* variants, possibly underestimating the number of causative variants responsible for dominant forms. Two patients, P32 and P173, presented the tibial muscular dystrophy (TMD) Iberian mutation. Interestingly, patient P173, carrying a splice mutation in addition of the Iberian mutation, had a more complex phenotype with earlier onset than patient P32 who carried a VUS missense variant in addition to the Iberian mutation. As Evilä A and colleagues stated [30], second titin mutations can explain atypical phenotypes, so the splicing mutation identified in patient P173 could explain his more severe phenotype in contrast with the non-truncating missense variant identified in P32.

Although we did not include patients with suspected dystrophinopathy in this study, we included *DMD* in the panel to detect DMD and BMD cases that may not have been suspected previously on the basis of clinical data. For this reason, despite being one of the most frequent muscular dystrophies in infancy, we only detected one patient with a BMD. The same applies to LGMDs as our laboratory performs routine genetic diagnosis for *DYSF*, *ANO5*, *LMNA*, *SGCA*, *SGCB*, *SGCG*, *SGCD*, *CAV3*, *FKRP*, *MYOT* and *DES* genes. Thus, the results of this work do not reflect the prevalence of the diseases caused by these genes in our population. 

Targeted NGS enables a rapid and cost-effective analysis of NMD-related genes, offering reliable results in a short time, reducing difficulties of storage and data analysis, and minimizing the problem of incidental findings. This tool has also proven useful to diagnose CM patients in adulthood when the disease has progressed and it is not easy to identify the initial symptoms. Moreover, NGS in some patients has relegated muscle biopsy to second tier of the diagnostic workflow, often avoiding an invasive process, which is of particular concern in children.

Muscle biopsy, nevertheless, remains a powerful and informative tool to prioritize and choose candidate variants once NGS data has been analyzed. One of the greatest difficulties that NGS has shown is the interpretation and classification of variants of uncertain significance. Muscle biopsy can also be useful in these cases as it allows RNA isolation so as to perform an in-depth analysis that could be key to solving particular cases. However, muscle biopsy is an invasive technique and alternative methods can be used to obtain a source of RNA. Cooper St et al. demonstrated the utility of MyoD gene-modified fibroblasts for research and diagnosis of human muscle disorders, reproducing protein deficiencies associated with different forms of muscular dystrophy and providing a renewable source of muscle-specific mRNA [31].

We would like to highlight the importance of family segregation studies as they add valuable information to the classification of VUS and provide additional evidence regarding the association of variants with the disease. Segregation studies let us know whether the identified mutations are “de novo” or inherited, and they also contribute to elucidating the inheritance pattern of the disease in the family. Factors such as variable expressivity and incomplete penetrance may complicate the interpretation of segregation analysis. This is the case of P55 that carried a *VCP* c.1202A > G missense variant. This variant, previously described at the LOVD database as a VUS, did not correlate with the patient’s clinical phenotype but fitted very well with the features observed in the muscle biopsy. The discrepancy occurred because his apparently healthy mother presented exactly the same variant in *VCP*. In principle, we would rule out this *VCP* variant as the cause of the disease, but since *VCP* variants are inherited in an autosomal-dominant manner with incomplete penetrance and phenotypic variability even within families [32,33], we prefer to gather more evidence before classifying the aforementioned variant. 

We also found difficulties in assessing the variants identified in P8, P24 and P146. They presented only a *CAPN3* heterozygous mutation after sequencing the whole coding *CAPN3* region and also performing an MLPA to detect a large deletion or duplication. A dominant form of *CAPN3* with a milder phenotype was recently described, but deeper analysis of our cases is needed as they do not share the same 21-bp deletion as Vissing et al.’s cases [34].

It is important to include all genetic, clinical, pathological, and functional data obtained into public mutation databases to improve our knowledge and interpretation of VUS. Sharing data among professionals is a key way to further knowledge concerning the genetic basis of neuromuscular diseases and improve their diagnosis and management.

High throughput sequencing technology is helping to reach high rates of genetic diagnosis, but certain limitations and difficulties have yet to be overcome. Although we were able to detect a large deletion in *MTM1* by NGS coverage analysis, this method is not reliable to detect copy number variations. Similarly, since targeted NGS is based on short read lengths, it is not an appropriate method to detect the molecular cause of some NMDs, such as repeat expansions in myotonic dystrophy or complex re-arrangements where exons remain intact. Another difficulty with NGS is the large amount of data generated. So much data implies the need to assess large numbers of identified variants and relate these to a patient’s clinical features and muscle pathology. Based on our experience, a great way to reach the most accurate diagnosis is to form multidisciplinary teams consisting of a neurologist/pediatric neurologist, a pathologist, and a geneticist who can jointly assess all the clinical and genetic data obtained in each family. Valuing all the data as a whole will be key to identifying causal mutation among the hundreds or thousands of variants identified. Preferably, prioritization of variants is based on a good genotype-phenotype correlation. However, in some cases a strong genetic candidate variant can lead to new diagnostic orientations. This is the case of patient P29 in our study, who was first suspected of a centronuclear myopathy. His sequencing data showed a known and reported pathogenic *COLQ* mutation, c.1289A > C (rs121908923) in a homozygous state, which could not be ignored. This genetic evidence led to clinical reassessment, finally establishing a definitive diagnosis of CMS. The patient was successfully treated with ephedrine, with a clear improvement in quality of life. Frequently, CMS can be misdiagnosed because their phenotype overlaps with that of congenital myopathies or muscular dystrophies. In the present work, we diagnosed 16 patients with CMS, although six of them were referred as atypical forms of muscular dystrophies or myopathies. Achieving an accurate genetic diagnosis in these patients is particularly important considering that their genetic defect will be crucial in determining the drug to be administered. For this reason, we strongly recommend the inclusion of CMS-related genes in gene panels developed to study myopathies. 

The use of NGS is becoming widespread in clinical laboratories. Its versatility, high performance, cost-effectiveness, and time efficiency make it a fine asset for studying genetic diseases with high clinical and genetic heterogeneity. Our custom next-generation sequencing panel proved to be a cost-efficient diagnostic strategy but we cannot overlook the fact that a definitive genetic diagnosis was not obtained for half of the patients. In response to the needs of these genetically undiagnosed patients, we will have to extend the genetic study to a greater number of genes, using, for instance, whole exome sequencing or whole genome sequencing. Additionally, it was described that ~25% of exonic mutations causing human inherited disease are likely to induce exon skipping either destroying conserved splicing enhancers or creating splicing silencers [35]. This data is particularly interesting for the assessment of the many missense mutations identified and also strengthens the relevance and importance of transcriptome analysis in mutation detection. 

## 5. Conclusions

In conclusion, our genetic diagnosis approach combined with clinical and histological data is useful and efficient to screen disease-causing mutations in patients with neuromuscular diseases. We consider that an integrated diagnosis performed by a multidisciplinary team with expertise in the neuromuscular field is the best way to evaluate and to choose the most appropriate technique and/or workflow for each particular case. 

## Figures and Tables

**Figure 1 genes-11-00539-f001:**
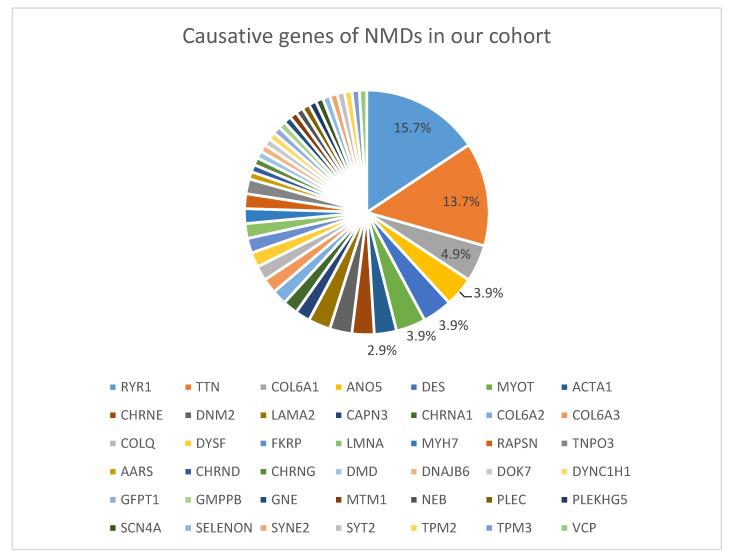
Causative genes of NMDs in our cohort. Percentages indicate the proportion of cases with the corresponding genetic defect among solved cases.

**Table 1 genes-11-00539-t001:** Genes included in the custom targeted next-generation sequencing panel.

*AARS*	*CRYAB*	*KLHL40*	*PREPL*
*ACTA1*	*DAG1*	*KLHL41*	*PTPLA*
*ACVR1*	*DES*	*KLHL9*	*PTRF*
*AGRN*	*DMD*	*LAMA2*	*RAPSN*
*ALG13*	*DNAJB6*	*LAMB2*	*RYR1*
*ALG14*	*DNM2*	*LAMP2*	*SCN4A*
*ALG2*	*DOK7*	*LARGE*	*SEPN1*
*ANO5*	*DPAGT1*	*LDB3*	*SGCA*
*B3GALNT2*	*DPM1*	*LMNA*	*SGCB*
*B3GNT1*	*DPM2*	*LMOD3*	*SGCD*
*BAG3*	*DPM3*	*LRP4*	*SGCG*
*BIN1*	*DYNC1H1*	*MEGF10*	*SGK196*
*CAPN3*	*DYSF*	*MSTN*	*SPEG*
*CAV3*	*EMD*	*MTM1*	*STIM1*
*CCDC78*	*FHL1*	*MTMR14*	*SYNE1*
*CFL2*	*FKRP*	*MUSK*	*SYNE2*
*CHAT*	*FKTN*	*MYBPC3*	*SYT2*
*CHKB*	*FLNC*	*MYF6*	*TCAP*
*CHRNA1*	*GAA*	*MYH2*	*TIA1*
*CHRNB1*	*GARS*	*MYH7*	*TMEM43*
*CHRND*	*GFPT1*	*MYO18B*	*TMEM5*
*CHRNE*	*GMPPB*	*MYOT*	*TNNT1*
*CHRNG*	*GNE*	*NEB*	*TNPO3*
*Cntn1*	*GTDC2*	*PABPN1*	*TPM2*
*COL12A1*	*HSPB8*	*PLEC*	*TPM3*
*COL6A1*	*IGHMBP2*	*PLEKHG5*	*TRIM32*
*COL6A2*	*ISPD*	*POMGNT1*	*TRPV4*
*COL6A3*	*ITGA7*	*POMT1*	*TTN*
*COLQ*	*KBTBD13*	*POMT2*	*VCP*

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
