# Peer review of "Targeted Next-Generation Sequencing in a Large Cohort of Genetically Undiagnosed Patients with Neuromuscular Disorders in Spain"

_genes, 2020, doi:10.3390/genes11050539_

Round 1

Reviewer 1 Report

It is a comprehensive study of Spanish patients with inherited neuromuscular disorders and genetic characterization of them. The manuscript is well written and the study data collection, results and discussion are well presented. I have no technical issues with the paper. I am not surprised that they found disproportionately larger number of RYR1 and TITIN gene mutations, as would be expected from an NGS study given its capability of capturing large genes such as RYR1 and TITIN (and Nebulin). The clinical correlation does suggest that these sequence variations are indeed pathogenic. However, not much in the way of biochemical studies or tissue based data (such as protein staining or histopathological data) is provided. I still think the paper is adequate without this additional data, but it would have made it even stronger. 

Author Response

Dear reviewer 1,

Thank you very much for your comments. We agree that it would have been preferable to introduce more biochemical and pathological data, but it has been a difficult issue since the samples came from several centres all around the country, and not all of them work in the same way nor do they collect the same data. At first, we considered including only those patients who had all the complete clinical, biochemical and pathological information, but on the other hand, we wanted to test the power of the panel in a real clinical context, where it is not always possible to collect all the information.

Anyway, we completely agree with reviewer's opinion. Having more data in more patients would have been very positive for the study.

Reviewer 2 Report

The Authors report the molecular results obtaining through the NGS technique in a group of 207 patients addressed to their lab for a suspicion of neuromuscular disorder. The NGS panel included 116 genes known to cause congenital myopathies, congenital and limb girdle muscular dystrophies, congenital myasthenic syndromes and non dystrophic myotonias.

A diagnosis was reached in about 50% of the patients included in the study, one third represented by central core myopathy (15,7%) and titinopathy (13,7).

The study is well conducted and confirm NGS technique as the best approach to reach a diagnosis in people often showing overlapping phenotypes, avoiding the use of more invasive procedure.  

I have no specific comments, except for one point that in my opinion needs to be clarified:

At page 6, lines 208-209, the AA say: “The most common causative genes identified were……….TTN (14 cases; 13.7%), while at the same page, line 222, they affirm: “Twenty-four patients showed TTN candidate variants: thirteen showed a single variant, and the eleven showed two variants”. The question is: What is the exact number of patients, 14 or 24?

Author Response

Dear reviewer 2,

We sincerely appreciate your helpful comments.

We agree with you, the paragraph was confusing. Lines 208-209 referred to those patients in whom we were able to confirm that the identified TTN variants were responsible for the patient’s pathology. In line 222 we referred to all those identified TTN variants that matched with the patients’ phenotype but we could confirm their causality only in 14 cases. In the remaining 10 cases we could not perform further analysis, such as familiar segregation, to confirm de genetic diagnosis of titinopathy.

Thanks for the comment. Please do not hesitate to tell us if you think the paragraph is still unclear.

Reviewer 3 Report

This manuscript describes a considerable amount of work and all mutations, VUS should be reported.

NGS was used to identify mutations in a cohort of over 200 patients with a variety of neuromuscular disorders.

The manuscript is well written clear and logical and while the authors almost seem apologetic/disappointed with the mutation detection rate, nearly 50% is very impressive.

I have some minor comments and suggestions that could  further improve a very nice piece of work.

Should there be a mention of mitochondrial disorders?

Line 78.  replace "many such" with most

Line 192.  mention of mRNA.  I assume this was possible from the muscle biopsies.  There should be details of methods.

Line 228.  perhaps "genetically diagnosed" could be placed with confirmed with a genetic diagnosis

Line 255while the P82 mutation is intronic, the position is well within the consensus donor splice site

Line 264. Missense mutations in the dystrophin gene are rare.  Perhaps some more details would be appropriate.

Line 353.  Perhaps this would be a position to introduce alternatives to muscle biopsies which are very invasive.  Could authors discuss use of skin fibroblasts and forced myogenesis.  This is not perfect but allows patient specific myogenic cells to be analysed from a skin punch.

Line 380. NGS will also miss complex gene re-arrangements (inversions, etc) where exon are left intact.

Line 409.  To account for missing gene mutations there are several obvious possibilities.  

As discussed , new genes to be included in the screen.

However, the "average" human gene consists of ~9 exons spanning 30kb for a 1kb mRNA.  There is a lot of missing intronic sequence in the screen so deep intronic mutations will not be detected.

Perhaps more importantly, there could be mention of a retrospective analysis of confirmed pathogenic nonsense and missense mutations showing a high proportion of mutations caused abnormal splicing (as high as 25%). Timothy Sterne-Weiler, Jonathan Howard, Matthew Mort, et al. Genome Res. 2011 21: 1563-1571.  Loss of exon identity is a common mechanism of human inherited disease.

Perhaps this could be mentioned in light of the many missense mutations identified.  Also strengthens the relevance and importance of transcriptome analysis in mutation detection.

Author Response

Dear reviewer 3,

Thanks for your comments and suggestions. We have made some changes that are detailed below:

“Should there be a mention of mitochondrial disorders?”.

We agree that a mention of mitochondrial disorders should be included. A new paragraph has been included: “Mitochondrial diseases partially overlap with clinical presentation of neuromuscular disorders. Despite mitochondriopathies constitute a heterogeneous group of diseases with widely varying clinical features, involvement of both muscle and nerve is quite common, due to its dependence on mitochondrial activity. Consequently, myopathy and neuropathy are a major, often presenting, feature of a number of mitochondrial syndromes. In addition, mitochondrial dysfunction may play a role in a number of classic neuromuscular diseases. However, clinical guidelines and protocols have been developed to discriminate between mitochondrial and neuromuscular disorders, that genetically rely on different aetiology”. It corresponds to lines 96-103. A brief sentence: “In addition, patients suspected of having mitochondrial disease were not included in this study” has also been included in lines 139, 140.   

“Line 78.  replace "many such" with most”. We have replaced it and now it corresponds to line 79.

“Line 192.  mention of mRNA.  I assume this was possible from the muscle biopsies.  There should be details of methods.”. Thank you for the observation; mRNA processing was missing in the methods section.  We have introduced a paragraph: “Messenger RNA analysis was performed by cDNA sequencing. Total mRNA was extracted and purified from approximately 30 mg of muscle using RNeasy Fibrous Tissue Mini Kit (Qiagen, Hilden, Germany) and subsequently retrotranscribed to cDNA by RT-PCR using polythymine primers (ThermoFisher Scientific, Waltham, Massachusetts, USA). Self-designed primers were used to amplify the cDNA region in which the intronic mutation was located including two 5’ and 3’ contiguous exons.” that corresponds to lines 179-184.

“Line 228.  perhaps "genetically diagnosed" could be placed with confirmed with a genetic diagnosis”. We agree with the reviewer’s comment and we have made the change. Now it is in line 249.

“Line 255 while the P82 mutation is intronic, the position is well within the consensus donor splice site”. We agree with the reviewer’s observation and we have remade the sentence: “All splicing variants were located at +1, -1 and +2 splice site positions, except for patients P82 and P118, who showed different intronic location variants.” Now it corresponds to line 274-275.

Line 264. Missense mutations in the dystrophin gene are rare.  Perhaps some more details would be appropriate. Following the reviewer’s comment, we have added more details about the case: “In one family, an adult male (P98) harboured a missense mutation located at the exon 6 of DMD gene, which contains an acting binding site. After this finding, the patient was re-examined showing a phenotype compatible with a Becker muscular dystrophy, that was confirmed by a reduction of dystrophin in the muscle biopsy immunostaining. DMD deletions and duplications were discarded by MLPA analysis. Now it is in lines 284-288.

“Line 353.  Perhaps this would be a position to introduce alternatives to muscle biopsies which are very invasive.  Could authors discuss use of skin fibroblasts and forced myogenesis.  This is not perfect but allows patient specific myogenic cells to be analysed from a skin punch.” We completely agree with the reviewer opinion. We have added the following sentence and a bibliographic reference: “However, muscle biopsy is an invasive technique and alternative methods can be used to obtain a source of RNA. Cooper St et al. demonstrated the utility of MyoD gene-modified fibroblasts for research and diagnosis of human muscle disorders, reproducing protein deficiencies associated with different forms of muscular dystrophy and providing a renewable source of muscle-specific mRNA [28].” It is in lines 376-380.

“Line 380. NGS will also miss complex gene re-arrangements (inversions, etc) where exon are left intact.” We have completed the sentence adding the reviewer’s observation: “Similarly, since targeted-NGS is based on short read lengths, it is not an appropriate method to detect the molecular cause of some NMDs, such as repeat expansions in myotonic dystrophy or complex re-arrangements where exons remain intact.” It is in line 408.

“Line 409.  To account for missing gene mutations there are several obvious possibilities.  

As discussed, new genes to be included in the screen.

However, the "average" human gene consists of ~9 exons spanning 30kb for a 1kb mRNA.  There is a lot of missing intronic sequence in the screen so deep intronic mutations will not be detected.

Perhaps more importantly, there could be mention of a retrospective analysis of confirmed pathogenic nonsense and missense mutations showing a high proportion of mutations caused abnormal splicing (as high as 25%). Timothy Sterne-Weiler, Jonathan Howard, Matthew Mort, et al. Genome Res. 2011 21: 1563-1571.  Loss of exon identity is a common mechanism of human inherited disease.

Perhaps this could be mentioned in light of the many missense mutations identified.  Also strengthens the relevance and importance of transcriptome analysis in mutation detection.”

We greatly appreciate this comment as well as the reference provided by the reviewer. We would like to request permission from the reviewer to use the phrase that he suggested in his comments: “in light of the many missense mutations identified.  Also strengthens the relevance and importance of transcriptome analysis in mutation detection”. We think that it perfectly defines the idea that we would like to express and we would like to use it if reviewer agrees. We have written this sentence in the manuscript and added the provided reference: “Additionally, it has been described that ~25% of exonic mutations causing human inherited disease are likely to induce exon skipping either destroying conserved splicing enhancers or creating splicing silencers [32]. This data is particularly interesting for the assessment of the many missense mutations identified and also strengthens the relevance and importance of transcriptome analysis in mutation detection.”. It is in lines 435-439.

Please do not hesitate to contact me if any further requirement is needed.

Yours sincerely,

Lidia GonzalezQuereda, PhD.

Genetics Dept. of Hospital de Sant Pau, Barcelona, Spain.

CIBERER, IIB Sant Pau

e-mail address: lgonzalezq@santpau.cat
